# Nocturnal Lifestyle Behaviours and Risk of Poor Sleep during Pregnancy

**DOI:** 10.3390/nu14112348

**Published:** 2022-06-04

**Authors:** Chee Wai Ku, Rachael Si Xuan Loo, Michelle Mei Ying Tiong, Sing Yee Clara Eng, Yin Bun Cheung, Lay See Ong, Kok Hian Tan, Mary Foong-Fong Chong, Jerry Kok Yen Chan, Fabian Yap, See Ling Loy

**Affiliations:** 1Department of Reproductive Medicine, KK Women’s and Children’s Hospital, Singapore 229899, Singapore; gmskcw@nus.edu.sg (C.W.K.); jerrychan@duke-nus.edu.sg (J.K.Y.C.); 2Duke-NUS Medical School, 8 College Road, Singapore 169857, Singapore; clara.eng@u.duke.nus.edu (S.Y.C.E.); laysee.ong@mohh.com.sg (L.S.O.); tan.kok.hian@singhealth.com.sg (K.H.T.); fabian.yap.k.p@singhealth.com.sg (F.Y.); 3Department of Paediatrics, KK Women’s and Children’s Hospital, Singapore 229899, Singapore; rachael.loo.s.x@kkh.com.sg; 4KK Women’s and Children’s Hospital, Singapore 229899, Singapore; michelle.tiong@mohh.com.sg; 5Centre for Quantitative Medicine, Duke-NUS Medical School, Singapore 169857, Singapore; yinbun.cheung@duke-nus.edu.sg; 6Program in Health Services & Systems Research, Duke-NUS Medical School, Singapore 169857, Singapore; 7Tampere Centre for Child, Adolescent and Maternal Health Research, Tampere University, 33014 Tampere, Finland; 8Department of Maternal Fetal Medicine, KK Women’s and Children’s Hospital, Singapore 229899, Singapore; 9Singapore Institute for Clinical Sciences, Agency for Science, Technology and Research (A*STAR), Singapore 117609, Singapore; mary_chong@nus.edu.sg; 10Saw Swee Hock School of Public Health, National University of Singapore, Singapore 117549, Singapore; 11Lee Kong Chian School of Medicine, Nanyang Technological University, Singapore 636921, Singapore

**Keywords:** circadian rhythm, light exposure, nocturnal eating, screen viewing, physical activity, pregnancy, sleep quality

## Abstract

The extent to which lifestyle practices at night influence sleep quality in pregnant women remains unknown. This study aimed to examine whether nocturnal behaviours were associated with poor sleep during pregnancy. We performed a cross-sectional analysis of a prospective cohort of pregnant women at 18–24 gestation weeks recruited from KK Women’s and Children’s Hospital, Singapore, between 2019 and 2021. Nocturnal behaviours were assessed with questionnaires, and sleep quality was measured using the Pittsburgh Sleep Quality Index (PSQI) with a global score ≥5 indicative of poor sleep quality. Modified Poisson regression and linear regression were used to examine the association between nocturnal behaviour and sleep quality. Of 299 women, 117 (39.1%) experienced poor sleep. In the covariate-adjusted analysis, poor sleep was observed in women with nocturnal eating (risk ratio 1.51; 95% confidence interval [CI] 1.12, 2.04) and nocturnal artificial light exposure (1.63; 1.24, 2.13). Similarly, nocturnal eating (β 0.68; 95% CI 0.03, 1.32) and light exposure (1.99; 1.04, 2.94) were associated with higher PSQI score. Nocturnal physical activity and screen viewing before bedtime were not associated with sleep quality. In conclusion, reducing nocturnal eating and light exposure at night could potentially improve sleep in pregnancy.

## 1. Introduction

The prevalence of poor sleep quality in the general population ranges between 10 and 48% [1,2,3]. Many women experience significant sleep disturbances during pregnancy [4]. A study reported that 45.7% of pregnant women experience poor sleep quality [5]. This can be the result of a multitude of biological and physiological changes [6], including sleep apnoea [7], increased micturition [8], and restless leg syndrome [9]. As pregnancy progresses, the frequency of these nocturnal awakenings increases, and sleep quality worsens [5]. Nocturnal lifestyle habits such as eating at late hours and leaving the lights on have been shown to affect sleep quality in the non-pregnant population. However, there is a lack of understanding of how they can affect sleep quality in expectant mothers. Given the widespread nature of poor sleep among pregnant women and its association with adverse pregnancy outcomes and poor quality of life [10], it is crucial to understand the contributing factors and provide guidance to expectant mothers to improve sleep quality.

Many nocturnal lifestyle habits have been associated with sleep quality [11]. According to the International Classification of Sleep Disorders, the main categories of nocturnal lifestyle habits that can result in poor sleep include an inconsistent sleep–wake schedule, an unconducive sleeping environment, the use of stimulants such as coffee or nicotine, and poor pre-sleep behaviours such as studying in bed [12]. Although some have postulated that sleep hygiene could be more a contributing factor than a causal factor [13], a few interventional studies have shown that sleep hygiene education leads to better sleep [14]. However, studies that explore the impact of lifestyle factors on sleep were mostly aimed at the general population [15,16]. Therefore, there is insufficient evidence to make recommendations for pregnant women. Our study aims to investigate which nocturnal behaviours are associated with poor sleep during pregnancy, which will guide the development of recommendations to improve sleep quality for pregnant women. 

## 2. Materials and Methods

### 2.1. Study Design and Participants

We performed a cross-sectional analysis using data from a prospective cohort study designed to examine maternal night-eating patterns and glucose tolerance during pregnancy (Clinicaltrials.gov, NCT 03803345), as described elsewhere [17]. Women receiving antenatal care between 18 and 24 weeks of gestation at KK Women’s and Children’s Hospital (KKH) in Singapore were recruited from March 2019 to October 2021. These women were at least 18 years old and Singaporean citizens or Singapore permanent residents. Women diagnosed with gestational diabetes mellitus (GDM) at recruitment, with pre-existing type-1 or type-2 diabetes, on routine night-shift work, or using anticonvulsant medications or oral steroids were excluded. All participants provided written informed consent at recruitment. This study was conducted according to the Helsinki Declaration and approved by the Centralised Institutional Review Board of SingHealth (reference 2018/2529).

### 2.2. Data Collection and Measures

At recruitment, trained research staff conducted a face-to-face interview with pregnant women between 18 and 24 weeks of gestation to determine maternal sociodemographic data and measured height using the SECA stadiometer (model 213, Hamburg, Germany) in the clinic. Participants were asked to recall their pre-pregnancy weight, which was used to determine pre-pregnancy body mass index (BMI) using the following formula: pre-pregnancy weight (in kg) divided by height (in meters) squared. Participants responded to questionnaires asking about activity levels in different periods of the day: ‘During which period(s) of the day do you usually perform moderate or vigorous activities: 7–11:59 a.m., 12–4:59 p.m., 5–6:59 p.m., 7–11:59 p.m., 12–4:59 a.m., 5–6:59 a.m.’. Moderate activity was defined as activity that required moderate physical effort and made an individual breathe somewhat harder than normal; vigorous activity was defined as activity that required hard physical effort and made an individual breathe much harder than normal. The use of electronic media before bedtime was assessed by asking ‘On average, how much time do you spend on electronic devices before going to sleep at night?’. Participants specified the number of days and duration per day or per week they viewed each of these devices: television, computer, tablet/mobile phone.

Artificial light exposure was assessed using the Harvard Light Exposure Assessment (H-LEA) questionnaire [18]. Participants self-reported the light source(s) to which they were exposed hourly for 24 h during a typical weekday (workday) and weekend (non-workday). In the H-LEA, light exposure was grouped into six types of light sources, with respective estimated corneal illuminance values (lux): darkness (0.2 lux), sunlight/outdoor natural light (2000 lux), indoor natural light (200 lux), fluorescent lamp (100 lux), halogen/incandescent light (20 lux), and other artificial light sources such as television or phone use in the dark (10 lux). The grouping was slightly different from the original version of H-LEA, which contained seven types of light sources [18]; we combined halogen and incandescent light sources into a single group, as both have the same corneal illuminance values (lux) and would be difficult to distinguish by participants. On the same H-LEA, we additionally asked the participants to specify at which hours throughout 24 h they have their meals during a typical weekday and weekend. 

Participants self-administered the Depression Anxiety Stress Scale (DASS21), which consists of 21 self-rated items on a Likert scale (0–3), grouped into three subscales (each contains seven items), assessing the severity of depressive, anxiety, and stress symptoms [19]. Participants with the presence of depression (score > 9), anxiety (score > 7), or stress (score > 14) were classified as having negative emotion.

Perceived sleep quality was assessed using the Pittsburgh Sleep Quality Index (PSQI) questionnaire [20]. The PSQI consisted of 19 items that were rated on a four-point Likert scale (0 to 3) and grouped into seven components that comprise subjective sleep quality, sleep latency, sleep duration, sleep efficiency, sleep disturbances, use of sleep medications, and daytime dysfunction. A global PSQI score was generated by summing the scores from these seven components (range 0 to 21), with <5 and ≥5 indicating good and poor sleep, respectively [20]. The PSQI has been shown to have similar psychometric properties in pregnant and non-pregnant populations [21,22].

### 2.3. Nocturnal Behaviours

Nocturnal activity was defined by moderate-to-vigorous activity performance after 7 p.m., which is approximately four hours before the average bedtime among pregnant women in Singapore [23], a range used by other studies to assess the effects of exercise at night on sleep [24]. Given that only five participants reported moderate-to-vigorous activity during 12:00–4:59 a.m., we combined this with activities done during 7:00–11:59 p.m. Total hours spent on screen before bedtime was computed from frequency and duration of viewing television, computers, and tablet/mobile phones in a week. Nocturnal screen time use was determined by screen viewing before bedtime at night for >1 h per day, given that cutting off screen time one hour before bed is recommended [25]. To estimate the hourly light exposure values, mean corneal illuminance was calculated from the H-LEA using estimated mean illuminance values (lux) for each light source during the weekday and weekend as indicated in the questionnaire [18]. Nocturnal artificial light exposure was defined based on the mean corneal illuminance with lux ≥ 5 between 2 and 4 a.m. Evidence showed that the peak of melatonin occurs during this period and its level could be suppressed by dim light with 5 lux [26,27,28,29]. To determine nocturnal eating, frequencies of meal intake from 8 p.m. to 4:59 a.m. during weekdays and the weekend were summed and classified into with and without meal intake during this period.

### 2.4. Statistical Analysis

Continuous data were presented as means and standard deviations (SDs); categorical data were presented as frequencies and percentages. Differences in maternal characteristics between women with good and poor sleep were compared using independent t-test for continuous variables and Pearson’s Chi-square test for categorical variables. We used modified Poisson regression [30] and linear regression to examine the association between nocturnal lifestyle behaviour and sleep quality as a dichotomy and quantitative variable, respectively. Nocturnal behaviours were mutually adjusted in a single model and further adjusted for maternal age, ethnicity, years of education, employment status, working overtime, and pre-pregnancy BMI. Given the potential role of emotion acting as a confounder or mediator in the association between nocturnal behaviours and sleep quality, we further adjusted it in the model. These covariates were identified from the literature review [5,31,32] and based on the directed acyclic graph. We subsequently tested interactions between nocturnal behaviour and pre-pregnancy BMI on sleep quality, by introducing each of the cross-product terms of nocturnal behaviour and BMI into the fully adjusted models. All statistical analyses were performed using Stata 16 (Stata, College Station, TX, USA).

## 3. Results

### 3.1. Participant Characteristics

In total, 299 pregnant women at a mean of 20.4 gestational weeks of were included in this study, after excluding one participant with incomplete data. The majority of the women were of Chinese ethnicity (81.9%), followed by Malay (16.1%), Indian (1.7%), and others (0.3%), which were grouped as non-Chinese. Of these women, 117 (39.1%) exhibited poor sleep, with an overall mean global PSQI score of 5.43 (SD 2.81). Table 1 presents maternal characteristics based on sleep quality. Women with poor sleep were more likely to be non-Chinese, to exhibit negative emotion, and to have higher pre-pregnancy BMI. Age, years of education, employment, and overtime work status did not differ significantly between women with good and poor sleep.

### 3.2. Associations between Nocturnal Behaviours and Sleep Quality

Higher percentages of poor sleep were observed in women with meal intake after 8 p.m. (56.4% vs. 33.5%; *p* < 0.001) and artificial light exposure ≥5 lux between 2–4 a.m. (21.4% vs. 6.0%; *p* < 0.001), but not in those with moderate–vigorous physical activity performance after 7 p.m. (19.7% vs. 22.5%; *p* = 0.555) and screen viewing before bedtime for >1 h (81.2% vs. 76.4%; *p* = 0.324) (Figure 1). Table 2 shows the associations between nocturnal behaviours and sleep quality. In the confounder-adjusted modified Poisson regression model (Model 3), nocturnal eating (risk ratio 1.51; 95% confidence interval [CI] 1.12, 2.04) and nocturnal artificial light exposure (1.63; 1.24, 2.13) were associated with a higher risk of poor sleep, after adjusting for sociodemographic characteristics and pre-pregnancy BMI. Further adjustment of negative emotion attenuated the associations of nocturnal eating (1.33; 1.00, 1.78) and nocturnal artificial light exposure (1.34; 1.02, 1.76) with poor sleep (Model 4). No associations were found for nocturnal physical activity (0.85; 0.61, 1.18) and screen viewing before bed (1.12; 0.77, 1.63) with sleep quality. Similar findings were observed for global PSQI score in the multiple linear regression models with covariate adjustment (Table 3). No interactions were observed between nocturnal behaviours and pre-pregnancy BMI in relation to sleep quality (all *p*-interaction > 0.500).

## 4. Discussion

Poor sleep quality is a common phenomenon in pregnancy. In this study, we found that nearly 40% of expectant mothers reported poor sleep quality between 18 and 24 weeks gestation. With the assessments of different nocturnal behaviours and their associations with sleep quality, we showed that meal intake after 8 p.m. and artificial light exposure at night were associated with poor sleep, after adjusting for baseline socio-demographics and pre-pregnancy weight. These associations were attenuated by emotion. On the contrary, physical activity and screen viewing before bedtime had no significant association with sleep quality. 

In pregnant women, nocturnal eating is common as they have physiological demands to meet an increased appetite [33]. Our study showed that nocturnal eating in pregnancy was associated with poor sleep quality, which is corroborated by previous studies in pregnant women [23,34]. Nocturnal eating of carbohydrate-rich foods, in particular, has paradoxical effects on sleep. On the one hand, it can decrease sleep latency [35] due to the elevation of tryptophan [36] and the suppression of orexin [37]. However, it also results in poorer sleep quality [38,39], as the circadian rhythm of core body temperature is delayed and nocturnal melatonin secretion is reduced [40]. In turn, this disruption of the circadian rhythm can lead to metabolic dysfunction [41], such as an increased risk of insulin resistance, predisposing to diabetes [42], as well as a higher chance of preterm birth [23,43]. However, strict restriction of nocturnal eating can also result in undue psychological stress and similarly lead to preterm birth [44]. As such, pregnant women might consider eating a small amount of food that is low in calories and rich in protein [45], serotonin, and folate, such as kiwifruit [46], or foods that contain melatonin, such as eggs, fish, nuts, and mushrooms [47], to address their physical needs while not compromising their sleep quality. Nevertheless, it is worth noting the potential presence of reverse causality, where poor sleep might contribute to nocturnal eating. Since this study did not examine the amount and breakdown of the type of food eaten and its association with sleep quality, as well as the exploration of reasons that led to pregnant women engaging in nocturnal eating, these could be studied in future research to guide recommendations.

Exposure to artificial light at night was also associated with poorer sleep quality during pregnancy. The pervasive use of nocturnal artificial light was associated with poorer health outcomes in the non-pregnant population [48,49], including an increased risk of cancer [50,51,52] and type 2 diabetes [53,54]. Light exposure causes a disturbance of the biological circadian rhythm due to a reduction in melatonin production, which is responsible for the regulation of the sleep–wake cycle [55], resulting in poorer sleep quality. In pregnancy, melatonin has been shown to be beneficial for the optimal functioning of the placenta and foetus [56,57] and is essential for a successful pregnancy. Reduced levels of melatonin have been implicated in pregnancy complications, including pre-eclampsia and neurological disabilities in neonates [57]. Therefore, reducing exposure to artificial light at night might help improve sleep quality in the pregnant population and lead to optimal levels of melatonin, which is crucial for improved maternal and foetal outcomes. It is recommended that the sleep environment be as dark as possible, with a maximum average illuminance value of 1 lux [58]. However, we cannot exclude the possibility that artificial light exposure at night could be the result of poor sleep instead. Well-controlled studies with more detailed investigations longitudinally (e.g., record of daily sleep diary, use of objective sleep and light tracker for a certain duration) are required to ascertain this cause–effect relationship.

Screen viewing before bed is a common habit in modern times [59]. About 80% of the participants in our study engaged in screen viewing before bed; however, this did not affect sleep quality. The nature of screen interaction and brightness of the screen, other than the act of screen viewing alone, might play greater roles in affecting sleep quality. Although white light LEDs found in most modern devices, such as televisions and smartphones, have peak emission of blue light ranges that can suppress melatonin secretion [60,61] and hence affect sleep quality, the intensity of the blue light remains a factor [60]. However, that was not measured in this study. Furthermore, the purpose of the technological devices used makes a difference, since interactive technological devices such as cell phones, laptops, and game consoles are more likely to result in poor sleep compared to passive ones such as television [59]. Future research should consider screen brightness, light emission, and content to better understand the impact of screen viewing on sleep quality.

Exercising prior to sleep does not result in poor sleep quality, which is consistent with findings available in the recent literature [24]. Traditionally, experts advise against exercising before bedtime, as it is thought to negatively affect sleep quality. The American Academy of Sleep Medicine 2001, among other studies, suggests that exercise before sleep can contribute to poor sleep quality [12]. However, more recent studies [24] sought to disprove this fact and suggested that exercising at least one hour before bed could, in fact, improve sleep quality. However, our study did not account for the absolute time interval between exercise and sleep. In addition, exercise during pregnancy has been shown to reduce the risk of preterm birth, since it promotes the development of the placenta [62]. Therefore, pregnant women are recommended to continue exercising during pregnancy, while more research is required to determine the benefits of exercising at least 1 h before bed.

This is one of the first studies to investigate the impact of nocturnal behaviours on sleep quality in pregnant women. The existing literature investigated the effect of nocturnal behaviours on sleep quality in the general population [15], with largely similar findings, so they are likely to be generalisable to pregnant women as well. However, the findings presented in this study may not be applicable to pregnant women at all gestation times, since this study was carried out between 18 and 24 weeks gestation. Sleep disturbances in the third trimester [63] have been well-described in the literature as a result of factors such as sleep apnoea and increased micturition. Being close to the equator, Singapore is not affected by seasonal changes, which can affect circadian cycles [64] and thus potentially influence nocturnal behaviours like nocturnal eating [65], hence making Singapore an ideal location to study nocturnal lifestyle behaviours in relation to sleep quality. A subjective measure of sleep quality was used to estimate sleep quality in the participants. However, objective measurement tools like polysomnography or actigraphy are less practical for use in large-scale studies. In addition, the assessment of nocturnal eating did not include the amount and breakdown of the type of food eaten, which may have an impact on sleep quality.

## 5. Conclusions

Our study showed that nocturnal eating and exposure to artificial light at night were associated with poor sleep quality in pregnant women between 18 and 24 weeks gestation, while screen viewing and exercising before sleep were not. Poor sleep quality is prevalent during pregnancy, with profound implications for maternal and foetal health. A reduction in nocturnal eating and minimising exposure to artificial light at night, especially between 2 and 4 am, are potential recommendations to be included in the maternal sleep hygiene guidelines during pregnancy. More studies are required to investigate whether interventions to modify these nocturnal behaviours will lead to improved sleep quality in pregnant women and improve pregnancy outcomes.

## Figures and Tables

**Figure 1 nutrients-14-02348-f001:**
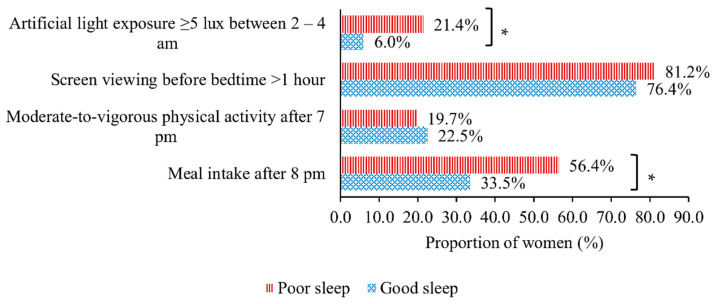
Bar chart illustrating the comparisons of women with good and poor sleep with respect to nocturnal behaviours (*n* = 299). * *p* < 0.001.

**Table 1 nutrients-14-02348-t001:** Maternal characteristics during pregnancy.

	Total (*n* = 299)	Good Sleep Quality (*n* = 182; 60.9%)	Poor Sleep Quality (*n* = 117; 39.1%)	*p*
Maternal Characteristics	Mean (SD)	Mean (SD)	Mean (SD)	
PSQI score	5.43 (2.81)	3.60 (1.20)	8.27 (2.15)	<0.001
Gestational age, weeks	20.36 (0.54)	20.34 (0.54)	20.38 (0.54)	0.557
Age, years	31.09 (4.24)	31.32 (4.28)	30.73 (4.17)	0.236
Education, years	14.29 (2.52)	14.38 (2.59)	14.15 (2.39)	0.430
Pre-pregnancy body mass index, kg/m^2^	22.89 (4.14)	22.22 (3.90)	23.92 (4.32)	<0.001
	*n* (%)	*n* (%)	*n* (%)	
Ethnicity				0.001
Chinese	245 (81.9)	160 (87.9)	85 (72.7)	
Non-Chinese	54 (18.1)	22 (12.1)	32 (27.4)	
Employment status				0.240
Unemployed	47 (15.7)	25 (13.7)	22 (18.0)	
Employed	252 (84.3)	157 (86.3)	95 (81.2)	
Working overtime				0.551
<3 times per week	255 (85.3)	157 (86.3)	98 (83.8)	
≥3 times per week	44 (14.7)	25 (13.7)	19 (16.2)	
Negative emotion				<0.001
No	168 (56.2)	128 (70.3)	40 (34.2)	
Yes	131 (43.8)	54 (29.7)	77 (65.8)	

Chi-square tests for categorical variables and independent-sample t tests for continuous variables were used to compare the two groups. SD, standard deviation; PSQI, Pittsburgh Sleep Quality Index.

**Table 2 nutrients-14-02348-t002:** The associations between nocturnal lifestyle behaviours and poor sleep quality among pregnant women (*n* = 299).

	Model 1 ^a^	Model 2 ^b^	Model 3 ^c^	Model 4 ^d^
Nocturnal Lifestyle Behaviour	RR (95% CI)	RR (95% CI)	RR (95% CI)	RR (95% CI)
Meal intake after 8 p.m.	1.75 (1.32, 2.33)	1.66 (1.25, 2.21)	1.51 (1.12, 2.04)	1.33 (1.00, 1.78)
Moderate-to-vigorous physical activity after 7 p.m.	0.90 (0.62, 1.29)	0.84 (0.60, 1.17)	0.85 (0.61, 1.18)	0.91 (0.67, 1.24)
Screen viewing >1 h before bedtime	1.20 (0.83, 1.74)	1.16 (0.80, 1.69)	1.12 (0.77, 1.63)	1.02 (0.70, 1.48)
Artificial light exposure ≥5 lux between 2 and 4 a.m.	1.99 (1.51, 2.61)	1.79 (1.36, 2.35)	1.63 (1.24, 2.13)	1.34 (1.02, 1.76)

Data were analysed using modified Poisson regression models. RR, risk ratio; CI, confidence interval. ^a^ Model 1: crude model. ^b^ Model 2: nocturnal lifestyle behaviours were mutually adjusted in a single regression model. ^c^ Model 3: Model 2 + maternal age, ethnicity, education, employment status, working overtime, and pre-pregnancy body mass index. ^d^ Model 4: Model 3 + negative emotion.

**Table 3 nutrients-14-02348-t003:** The associations between nocturnal lifestyle behaviours and global Pittsburgh Sleep Quality Index score among pregnant women (*n* = 299).

	Model 1 ^a^	Model 2 ^b^	Model 3 ^c^	Model 4 ^d^
Nocturnal Lifestyle Behaviour	β (95% CI)	β (95% CI)	β (95% CI)	β (95% CI)
Meal intake after 8 p.m.	1.21 (0.58, 1.85)	1.03 (0.41, 1.65)	0.68 (0.03, 1.32)	0.36 (−0.24, 0.97)
Moderate-to-vigorous physical activity after 7 p.m.	0.07 (−0.71, 0.85)	−0.11 (−0.85, 0.64)	−0.16 (−0.90, 0.58)	−0.01 (−0.70, 0.68)
Screen viewing >1 h before bedtime	0.33 (−0.44, 1.11)	0.27 (−0.47, 1.01)	0.24 (−0.51, 0.99)	−0.01 (−0.70, 0.69)
Artificial light exposure ≥5 lux between 2 and 4 a.m.	2.45 (1.51, 3.39)	2.22 (1.27, 3.17)	1.99 (1.04, 2.94)	1.51 (0.61, 2.41)

Data were analysed using modified Poisson regression models. CI, confidence interval. ^a^ Model 1: crude model. ^b^ Model 2: nocturnal lifestyle behaviours were mutually adjusted in a single regression model. ^c^ Model 3: Model 2 + maternal age, ethnicity, education, employment status, working overtime, and pre-pregnancy body mass index. ^d^ Model 4: Model 3 + negative emotion.

## Data Availability

The data presented in this study are available within the article.

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
