# Peer review of "Nocturnal Lifestyle Behaviours and Risk of Poor Sleep during Pregnancy"

_nutrients, 2022, doi:10.3390/nu14112348_

Round 1

Reviewer 1 Report

The manuscript entitled "Nocturnal lifestyle behaviours and risk of poor sleep during pregnancy" by C. W. Ku et al. is one of the first studies to investigate the impact of nocturnal behaviors on the sleep quality of pregnant women. And for this reason only, this work is original and I reckon will be of high interest to the readers of Nutrients.

Apart from the originality, this work has been carried out with a sufficient number of pregnant women and by employing appropriate and well-established methodological approaches. The methods are provided sufficiently, especially with regard to the main sleep quality assessment which was carried out using the Pittsburgh Sleep Quality Index. Statistical analysis of the results has been carried out and results are presented in a clear fashion. Importantly, the conclusions of this work are supported by the measured metrics to a large extent. It has to be noted that the finding that exercise prior to bedtime has a low impact on sleep quality is quite interesting.

Taken together, the originality, scientific soundness and potential interest of this work to the readers, I see high merit and I suggest publication of this manuscript to Nutrients. 

Apart from some minor grammatical issues that can be revised, I have no other suggestions for improvements.

Author Response

We thank the reviewer for the positive comments and we have reviewed the manuscript for grammar. 

Reviewer 2 Report

Interesting and novel study.  There is currently a lack of studies on the important topic of sleep during pregnancy.

Comments:

- Association is not causation.  The authors are right to comment that "we cannot exclude the possibility that artificial light exposure at night could be the result of poor sleep" (234-6).  This might also be the case with nocturnal eating, and might be added in the Discussion.  Both behaviors might be linked to anxiety: some people feel safer with some light, and some people think night-time eating offers advantages for the pregnancy or the fetus, eg to improve fetal movements or fetal growth etc.  Anxiety might lead to poorer sleep, more light and more nighttime eating.

- Natural/artificial light exposure in most countries is heavily dependent on the season.  Equatorial Singapore may be the 'right' area to minimise such seasonal changes, and the authors should see this as a strength of the current study.  When there is little (or no) natural light, people tend to eat more at 'nighttime'.  Social factors become more prominent drivers of eating and sleeping, which is the basis of some many winter parties and the "winter fat" in northern areas.  Pregnant women obviously participate in these social activities.  One wonders whether there is a seasonal change in nighttime eating in Singapore. 
